# Framework for Screening and Evaluating the Competencies and Qualities of the Board of Directors in South Africa's State-Owned Companies

Modi Hlobo *, Tankiso Moloi  and Benjamin Marx

Department of Accountancy, University of Johannesburg, Johannesburg 2092, South Africa
* Correspondence: modih@uj.ac.za

**Abstract:** Purpose—This research paper presents a framework for screening and evaluating the competencies and qualities of the board of directors in South African state-owned companies (SOCs). Design/methodology/approach—This study conducted a systematic literature review to gather primary data which was used to prepare a questionnaire for two rounds of the Delphi process, where data was analysed both qualitatively and quantitatively. Findings—The findings from the study revealed the ideal competencies and qualities of individual directors, the optimal collective competencies of directors, and the most appropriate screening and evaluation methods that could be adopted to benefit SOCs. Originality/value—This paper adds to the limited studies investigating the competencies and qualities of directors in SOCs, as most research is focused on listed private companies. Furthermore, there is currently no framework in South Africa that outlines the process for screening and evaluating the competencies and qualities of directors in South Africa's SOCs. In an effort to support the South African government screen and evaluate the key competencies and qualities of directors in state-owned companies, this team has developed a theoretically informed framework that can be used to screen potential board members' abilities and capabilities before they are appointed as well as to evaluate the relevance of existing board members.

**Keywords:** board members; competencies; qualities; corporate failures; state-owned companies

## 1. Background

There are currently 715 state-owned entities (SOEs) in South Africa contributing approximately 27% of South Africa's R1 trillion gross domestic product (Kikeri 2016). These SOEs include state-owned companies (SOCs), which are governed by the Companies Act, and the subject of this study (Kikeri 2016). The 2016 Department of Public of Public Enterprise (DPE) Budget Vote revealed that the SOCs that the DPE oversaw were worth R908 billion and they employed more than 114,000 workers. Therefore, this study is focused on the competencies and qualities of directors charged with governance of SOCs.

Notwithstanding being entrusted with a critical role in developing the South African economy, these SOCs have undermined corporate governance values in recent years, as evidenced by their poor performance (Thomas 2012). According to the report of the Auditor General for 2017–2018, the audits of four of the SOCs overseen by the DPE were not completed by the statutory closing date since these SOCs were unable to show that they were "going concerns" (Auditor General 2018). Furthermore, these companies disclosed R1.9 billion in irregular expenditures (Auditor General 2018).

The challenges at SOCs are severe, and in order for these SOCs to remain financially sustainable and meet their operational obligations, the National Treasury is compelled to bail them out (Omarjee 2017). Unfortunately, the financial bailouts of these SOCs come at the expense of public service delivery, as the funds allocated for service delivery are then used to bail out these SOCs. In addition to their financial difficulties, these SOCs also struggle with operational and governance issues, such as misrepresentation of qualifications

by a board member (Public Protector 2014); the senior executive positions being vacant for a considerable amount of time (Kanyane and Sausi 2015); disputes among board members, as well as a high turnover rate among board members (McGregor 2011); and the failure of board members to hold a meeting board meetings due to a lack of quorum as a result of the resignation or suspension of some of the board members (Daily Maverick 2019).

These inadequacies in the governance of SOCs have spurred citizens to question whether the boards of directors are competent enough to create long-term value in these companies (McGregor 2011). Consequently, researchers have investigated director competencies in SOCs and have determined that incompetent board of directors and executives contribute to poor corporate governance and consequently poor performance (Kanyane and Sausi 2015)

In order to address these challenges, it is important to re-examine the way directors are screened and appointed and also how the existing directors' competencies and qualities are evaluated. Therefore, the study sought to develop a framework that would focus on the screening and evaluation of the qualities and level of competence of the board of directors in SOCs.

## 2. The Impact of Incompetent Directors and Their Unethical Conduct on Corporate Failures

Following the 2008–2009 corporate scandals that involved companies such as Enron, WorldCom, Lehman Brothers, and Parmalat, corporate governance practices have created an interest on the impact of directors' competence on corporate scandals (Yaser and Denise 2012). According to Yaser and Denise (2012), managers' and directors' collusion in greed, corrupt practices, fraud, embezzlement, and mismanagement of company resources led to these scandals.

Chatzkel (2003) identifies the lack of independence between the executive board and the non-executive board of directors and inadequate oversight by non-executives as the primary causes of Enron's demise. According to Liesman et al. (2002), most corporate failures are caused by excessive executive management control, incompetent boards, and passive investors.

Similar corporate scandals have occurred in South Africa. PPC, Regal Treasury African Fund, Fidentia, JCI-Randgold, and Macmed are among the country's most prominent scandals (Nag 2015). In 2017, South Africa witnessed the demise of Steinhoff, which has been described as the worst corporate disaster to ever strike South Africa (Lepule 2017; Motau 2018). According to Lepule (2017), the Steinhoff scandal cost R200 billion in total losses due to overstated revenue in subsidiaries and the creation of off-balance-sheet companies to conceal losses, manipulate earnings, and defraud taxes. Another recent scandal was the 2018 Tongaat Hulett scandal, in which the company's financial results had been overstated by approximately R4.5 billion (Lepule 2017). VBS Mutual Bank represents another major corporate failure in South Africa, with the bank directors, senior executives, and well-connected politicians defrauding nearly R2 billion in taxpayer money (Lepule 2017).

South Africa's SOCs have also been challenged by corporate scandals and have been plagued by state capture, which is estimated to have cost the country R4.9 trillion and missed opportunities (Merten 2019). According to Madumi (2018), the primary governance challenges that SOCs faces are incompetence and misconduct among executives and boards of directors. Thabane and Snyman-Van Deventer (2018) agree with Madumi (2018) and attribute SOC weaknesses to ineffective, generally politically appointed boards of directors. Thabane and Snyman-Van Deventer (2018) agree with Madumi (2018) and attribute SOC weaknesses to incompetent board members that are politically appointed.

The challenges facing South Africas SOCs are a source of concern among South Africans, including prominent people, judiciary, and leaders, who have publicly expressed their dissatisfaction with the entities' operations. Following the state capture commission, the report was issued and states that:

> With regard to the appointment of members of boards of directors of SOEs as
> well as senior executives, the commission found that this responsibility can no
> longer be left exclusively in the hands of politicians, as they have miserably
> failed in their constitutional mandate to lead these institutions successfully. It
> has been recommended that a body be established that would be entrusted to
> identify, recruit and select competent people for such appointments in SOEs.
> (State Capture Report 2022)

South Africas presidenct, President Cyril Ramaphosa admitted at the Commission of
Inquiry into state capture that:

> The parlous condition of state-owned enterprises was the result of a massive
> system failure in how the boards of SOEs were appointed, some of the (failures)
> may have been inadvertent, and some may have been purposeful. Some of (the
> appointments) were hidden and masked.

Therefore, the core problems faced by SOCs are, firstly, SOCs are managed by incompetent
political appointees, indicating insufficient screening of prospective board members. Secondly,
SOCs boards of directors are often constrained by political interference (Public Protector State
Capture Report, 2016/2017). Thirdly, governance and performance of SOC are not as well
researched as JSE-listed companies (Menozzi and Vannoni 2014); as a result, this has prevented
SOCs from identifying and implementing governance structures and procedures that could
prevent poor performance and financial mismanagement (Menozzi and Vannoni 2014). It is
therefore at the back of this background that this study developed the FSECQ to help screen
potential board members and continuously evaluate the relevancy of existing board members.

### 3. Research Objectives

Given the aforementioned issues, the primary purpose of this research is to design the
framework for screening and evaluating the competencies and qualities (FSECQ) of South
African SOC boards of directors.

### 4. Theoretical Review

Several theories are relevant in explaining the corporate governance principles of the
board of directors. The primary theories adopted in this study are agency theory, resource
dependency theory, human capital theory, social capital theory, and trait theory, as these
theories are associated with the competencies and qualities of directors.

- Agency theory: This theory is focused on the relation between the principals (the
  business owners) and the agents (represented by the managers). The shareholders
  appoint agents to manage the company and increase shareholder value (Boshkoska
  2015). Managers generally have excellent knowledge and expertise in the company's
  day-to-day operations. However, sometimes they act in their own self-interests rather
  than the shareholders' (owners') interests (Olowosegun and Moloi 2021; Mbanyele
  2020). Managers' need for self-interest undermines trust between shareholders and
  managers. As a result, shareholders are compelled to appoint a board of directors to
  oversee and protect their interests (Moloi and Marwala 2020; Mizrachi 2004).
  Directors are expected to be knowledgeable in various fields, including accounting, tech-
  nology, communications, and public policy, to oversee these managers effectively. As a
  result, boards of directors are responsible for closing the gap between shareholders' and
  management's interests and making critical decisions that benefit shareholders (Petra 2005).
  As with the private sector, SOCs are operated by executive managers, while directors
  have an oversight role. The agency theory is complicated in the SOC environment
  as the executives and directors do not own the companies. To further exacerbate the
  problem, the ministers are assigned to serve as the shareholder representative, yet they
  also do not own the SOCs.
  The real owners of the SOCs are the citizens of the country. This structure creates
  a conflict of interest among managers, directors, and shareholders (Thabane and

Snyman-Van Deventer 2018). According to Menozzi and Vannoni (2014), there is a "double agency" problem caused by conflicts between managers, boards of directors, politicians, and citizens.

In this double-agency phenomenon, neither the executive, the directors, nor even the ministers can be expected to protect the interests of the SOCs with the same dedication as they would have done if the SOCs had been their own companies. What intensifies this double agency problem is that citizens (who are shareholders) typically do not have the knowledge or institutional ability to monitor and evaluate the performance of directors and ministers (Thabane and Snyman-Van Deventer 2018). Thabane and Snyman-Van Deventer (2018) further reveal that an agency system of this nature enables ministers (who are supposed to represent the shareholders) to pursue their own interests instead of those of the citizens (Thabane and Snyman-Van Deventer 2018).

In accordance with the agency theory, the purpose of the boards is to monitor managers' activities and, more specifically, to use their expertise to advise the management of the company (Mbanyele 2020). Accordingly, the agency theory is relevant to the present study, which investigates the competence and quality of directors, who both monitor and advise management to ensure that they maximise shareholders' value.

- The Resource Dependency theory: The resource dependence theory of corporate governance is another theory adopted in this study. It holds that directors bring valuable resources to the organisation, notably knowledge, experience, and access to critical external contacts such as suppliers, buyers, and investors (Hillman et al. 2002). Therefore, it is evident that to be able to play their role efficiently, the board of directors needs to be resourceful and bring knowledge, skills, and networks to the companies they serve. This theory is applicable to the present study, which investigates the required competencies and qualities of the board of directors in SOCs.

- The Human Capital Theory: The board's role is not limited to monitoring management activities, as described in the agency theory, or providing essential resources and connecting the company with external resources, as highlighted by the resource dependence theory. The board provides other essential resources to the company (Hillman et al. 2002). Hillman and Dalziel (2003) refer to the ability of boards to provide the organisation with essential resources as "board capital", where the "human capital" of the board of directors is one of the most critical parts that the board of directors brings to the company (Jimenez et al. 2012).

    Over the years, there has been increasing empirical evidence supporting the positive impact of board human capital on firms' performance. The human capital theory focuses on individual directors' skills and experience in the boardroom. Reed and Wolniak (2005) identify directors' qualifications and experience as core aspects of the human capital theory. Johnson et al. (2013) consider essential elements of human capital to be business expertise, experience as a CEO, financial expertise, and boardroom experience. In addition, Kiel and Nicholson (2003) consider human capital as knowledge and skills acquired by the board of directors, such as operational skills, industry expertise, boardroom skills, and organisational-specific knowledge. These authors argue that when individual directors use their knowledge, skills, and talents, boards are more effective in carrying out their tasks (Johnson et al. 2013).

    These studies show no specific category of skill or knowledge with extraordinary benefits and that there are several advantages, weaknesses, and inconclusive outcomes within the human capital theory (Johnson et al. 2013). Although the studies above are inconclusive, the present study considers the human-capital theory to be valid in that it enables the board of directors to be constituted of members with the necessary combination of professional skills to enhance strategic decision making.

- The Social Capital Theory: Social capital can be defined as an interpersonal relationship between individuals, both inside and outside the company. According to Kim and Cannella (2008), the boardroom's social capital is perceived as an asset that provides the company with external relations and resources. The definition of social capital

depends primarily on whether the board is focused on external relations (external social capital) or internal relations (internal social capital) (Adler and Kwon 2002). External relations relate to ties that involve others outside the organisation, whereas internal ties include interpersonal relationships between directors. Kim and Cannella (2008) refer to external social capital or "bridging" forms of social capital and internal social capital or "bonding" forms of social capital.

Strong internal social capital amongst the board enhances trust between directors, facilitates the exchange of relevant information and knowledge, and improves decision making (Kim and Cannella 2008; Radin and Stevenson 2006). Because of these benefits, the present study adopted both the internal and the external social capital and considered both concepts useful for an effective board.

- Traits Theory: Over the years, researchers have paid close attention to the characteristics and qualities of good leaders, including their personality traits such as their motives, values, cognitive abilities, social as well as problem-solving skills, and expertise (Zaccaro 2007). Carmeli (2006) further reported that knowledge, abilities, and social skills are the most critical traits of board members. Epstein and Roy (2004) believe that integrity and moral values are essential qualities for board members and that such attributes contribute to positive results for the entire organisation, including its investors. The traits theory suggests that essential traits such as integrity, ethical values, and problem-solving are essential to a strong leader.

Following the above discussion on the five theories that underpin this study, the subsection that follows provides a literature review on the competencies and qualities of directors. This is followed by a literature review on screening and evaluating the competencies and qualities of directors.

## 5. Literature Supporting the Research Study

The literature review in this study systematically investigates the required competencies and qualities of directors as well as the methods used to screen and evaluate these directors.

### 5.1. The Competencies and Qualities of Directors

This study focuses on the competencies and traits required of SOC directors in South Africa. Competency is defined by Jokinen (2005) and Garratt (2008) as "the unique abilities, knowledge, and experience that enable a director to execute their assigned duty". According to San Lam (2013), experience and credentials are essential for determining competency. This research study focuses on the following abilities:

A.  *Educational qualifications*: According to Westphal and Milton (2000), board members with relevant qualifications contribute inventive ideas, and distinctive views that assist organisations in addressing complicated challenges. Therefore, board members with relevant educational qualifications bring valuable perspectives into the boardroom, which aids in policy development and supports ethical decision-making (Westphal and Milton 2000).

B.  *Political acumen:* The 2015 OECD guideline on the corporate governance of SOCs suggests that it is necessary for SOC boards to have board members with political expertise as these board members' understanding of SOCs and government would aid in establishing credibility with the shareholders. Obviously, there are concerns that such board members would politicise the board's operations even further. Therefore, it is important to recruit independent board members with political expertise who can add value to the SOCs (OECD 2015).

C.  *Experience in board positions:* Stone and Tudor (2005) state that the experience of directors, including experience in managing a business and serving as a board member, is crucial to the company's success. Westphal and Milton (2000) add that seasoned directors establish relationships more readily with other directors and industry players.

D.    *Sound Strategic Communication Skills Sound Interpersonal and Relational Skills:* Forbes and Milliken (1999) emphasise the importance of board members having personal and cognitive skills. These skills enable directors to communicate well with other board members and the company's management teams.

E.    *Critical Thinking:* The OECD 2014 Competency Framework describes critical thinking as 'a type of analytical thinking that allows people to identify fundamental issues in complex situations. As a result, critical analytical skills include the capacity to form a broad picture of the organisation and its operations and its competitive advantage and challenges, industry trends, and market prospects' (OECD 2014). This skill is, therefore, relevant for a director to have.

Although there is scant research on the personal traits and values of board members in SOEs, Yusoff and Alhaji (2012) believe that the ideal board should possess trustworthiness, exceptional communication, self-confidence, and a feeling of commitment. De Vries and Florent-Treacy (2002) observed that the analytical capabilities, relationship skills, and dedication of board members were essential qualities. Integrity and ethical principles are among the researched personal characteristics and values in this study.

F.    *Integrity and Ethical values*: According to Blake (2016), the essential qualities demanded of directors are honesty and ethics. As per Yukl (2008), a person's integrity is reflected in their honesty, ethical behavior, and dependability. Likewise, Korn-Ferry (2003) discovered that moral behavior, integrity, and honesty were the most important values for directors, followed by accomplishing outcomes, customer orientation, teamwork, dedication, and respect for others. Schwartz et al. (2005) assert that board members set the moral norm in the organizations they serve; hence, they must maintain a high degree of ethical conduct to ensure the success of the business.

Using a systematic literature review and two rounds of Delphi process the authors constructed Table 1 containing the following list of required directors' competencies and qualities as follows.

**Table 1.** Comprehensive list of Competencies and qualities required from a director serving in an SOC.

| **Competencies Critical for Individual Directors** |
| --- |
| Educational qualification (Undergraduate and Post Graduate) |
| Sound strategic communicating skills |
| Sound strategic interpersonal relations skills |
| Understand the strategic purpose of the organisation and its role in the country's development |
| Good ethical standards and understanding the importance of managing public resources |
| Critical Thinking |
| **Qualities and Values Critical for Individual Directors** |
| Ethical, and acting with integrity |
| No tolerance for ethical violations |
| Always fair, just, and not biased |
| Respect for fellow board members and executives |
| Kind and acting in a manner that is always beneficial to the team |
| Honest, reliable, and trustworthy |
| Encourage executives and employees to thrive, flourish, and develop innovative ideas |
| Walk the talk of ethical values |
| **Directors Demographics** |
| Age (balance of all ages) |
| Gender (50:50) |
| Race (represents the skills needed despite the racial background of the candidate) |

Source: Researcher's Own Conceptualisation.

### 5.2. The Screening Processes for Directors

In 2018, the OECD surveyed "board nominations processes" of SOEs in various countries and noted that in countries that had adopted the centralised form of ownership in SOEs, such as in Chile, Slovenia, and South Africa, the political Minister was responsible for nominating members to SOE boards. Whereas in countries where they had adopted the decentralised form of ownership, such as Brazil, Estonia, Latvia and Turkey, the line ministries were more often responsible for nominations. The public finance ministry was also allowed to appoint one or more representatives to the board (OECD 2012).

Following the results of its study, the OECD (2012) advised that ministerial decisions on board nominations be subject to some form of agreement by a larger group of ministers, the Cabinet, or the Head of State. This is true in several nations, such as in Norway and Sweden. In addition, the guidelines recommend that appointing board members should be supported by transparent and consistent mechanisms for nominating candidates to the board and should involve current members of the SOE board and non-government shareholders.

South Africa's legislature and guidelines are inconsistent when it comes to board normination. For instance, the Public Finance Management Act (PFMA) is silent and only stipulates circumstances when board members can be disqualified and removed as board members. Whilst King IV requires transparency in the board selection process, even though the Minister makes the final decision. This inconsistency deems it necessary to have the FSECQ which will prescribe an ideal screening method for potential board candidates.

### 5.3. The Evaluation Processes for Directors

The FSECQ also prescribed a framework for evaluating board members. Board evaluations are regarded as a useful governance practice since they help to determine the current state of the board's overall functioning and indicate any gaps that may be filled through future appointments. Board evaluation practices vary in different countries from informal evaluations conducted by the Chair to external experts and facilitators, formal self-evaluations, and board committees.

Board evaluations assist in assessing and enhancing board performance and offer the Chair and ownership function with useful information regarding potential board composition changes and future recruitment. Board evaluations assist in assessing and enhancing board performance and offer the Chair and ownership function with useful information regarding potential board composition changes and future recruitment.

According to the OECD survey (2018), in "board nominations processes" of SOEs in countries such as India, Sweden, and Vietnam, the results of the evaluation process influenced the nomination process by identifying necessary competencies and board member profiles.

In India, for instance, the board evaluation outcomes are reported in the Annual Performance Evaluation of the Administrative Ministry. In Sweden, the recommendations of the chair influence the board nomination process. In Vietnam, board members must provide self-evaluations to the ministers in charge of nomination and appointment. Consequently, it is clear that evaluation results play a significant role in re-nomination or discipline measurement (OECD 2012).

Even with board evaluation processes, South Africa's legislation provides no guidance on board evaluation. Both the Company's Act and the PFMA are silent on board evaluations and only stipulates circumstances when board members can be disqualified and removed as board members.

According to the PFMA and King IV, the relevant political ministries govern the SOCs, and they are also responsible for appointing and dismissing the board of directors and the CEO. Rossouw and Reddy (2009) and Williams (2010) note that because ministers appoint directors and executives, this system is susceptible to political appointments in which appointed individuals will favor ministers in their strategic decision-making.

Furthermore, it is worth noting that the Companies Act of 2008 and the PFMA of 1999 have long since been updated and that there has been a lot of developments in the corporate governance environment that need the legislature to be revised, particularly since such revisions could close the gaps that have caused some of the corporate failures in South Africa.

It is against this background that it becomes critical to have a framework that prescribes how directors' competencies and qualities should be screened before a director is appointed to the board of directors in and SOC.

## 6. Methodology

According to Belkhir (2009), Dey and Chauhan (2009), and Bhagat and Bolton (2008), the majority of research in corporate governance investigates the relationship between the board of directors and the financial success of corporations, and hence focuses on quantitative approaches. These studies disregard the other inherent features and characteristics of the board of directors that also affect the performance of companies (Leblanc and Gillies 2005; Korac-Kakabadse et al. 2001). Hunicke et al. (2004) undertook a qualitative study and conducted personal interviews and observations to evaluate the most influential director traits on the performance of an organisation.

Similarly, this study used a mixed-methods approach, including systematic literature analysis and the Delphi method, to formulate the FSEQC framework.

*Research Data Collection Methods*

Vandiar (2015) analysed JSE listed company directors and used a qualitative methodology where he first conducted a document analysis and used the results to compile questionnaires. Serretta et al. (2009) investigated core corporate governance dilemmas facing boards, and the investigation was qualitative and used the Delphi technique. This study adopted a similar research approach and used two rounds of the Delphi technique.

The Delphi technique is a qualitative research method used when obtaining consensus among a group of experts. It relies on the fact that several heads are better than one when making subjective speculations and that these experts will most likely make presumptions based on logical judgments instead of guessing (Weaver 1971).

The questionnaires used in the Delphi process were prepared using information collected from the review of the literature on processes of screening and evaluating directors, as well as the information on the competencies and qualities of directors.

In first round of the Delphi, the participants were requested to confirm the identified competencies and qualities and in the second round experts were required to rank the confirmed competencies and qualities and a consensus was reached.

## 7. Discussion and Findings

The primary objective of this study was to develop the FSECQ of directors in South Africa's SOCs. To achieve this objective, the Delphi technique was used.

In the first round of the Delphi process, the questionnaire required experts to confirm whether the competencies and qualities identified from the literature were a pre-requisite or were acquirable.

Table 2 presents the proposed FSECQ. It is envisaged that the proposed FSECQ will help SOCs to determine whether they accept a potential board candidate and to establish whether the existing board of directors has the required skills.

**Table 2.** Proposed framework of competencies and qualities for screening the potential individual board member of an SOC.

| Category | Quality or Competence | Detailed Description | Score |
|---|---|---|---|
| Non-negotiable qualities and competencies for individual directors | Competencies | Educational qualification (In General) | 5 |
| | | Sound strategic communicating Skills | |
| | | Sound Strategic Interpersonal Relations skills | |
| | | Understand the purpose of the organisation and its role in the country's development | |
| | | Good ethical standards and understanding the importance of managing public resources | |
| | | Critical thinking | |
| | Qualities | Ethical, and acting with integrity | |
| | | No tolerance for ethical violations | |
| | | Always fair, just, and not biased | |
| | | Respect of fellow board members and executives | |
| | | Honest, reliable, and trustworthy | |
| | | Encourage executives and employees to thrive, flourish, and develop innovative ideas | |
| | | Walk the talk of ethical values | |
| Very important qualities and competencies for individual directors | Competencies | Educational qualification (Undergraduate) | 4 |
| | Qualities | Regularly discuss high ethical standards | |
| Important qualities and competencies for individual directors | Competencies | Educational qualification (Post Graduate) | 3 |
| | | Political Acumen/Awareness | |
| | | Experience in serving in other board positions | |
| | Qualities | Directors should have representation of all ages | |
| | | Director's Race is not critical instead, skills needed in the board is more important | |
| Negotiable qualities and competencies of directors | | Kind and acting in a manner that is always beneficial to the team. | 2 |
| Nice to have qualities and competencies of directors | | 50:50 Gender representation in the board | 1 |

Source: Researcher's Conceptualisation.

The FSECQ is also divided into two parts: a section for screening the individual director competencies and qualities and then a second section developed to evaluate the competencies of the existing board of directors as a collective. The FSECQ has five (5) columns.

The first column deals with the category of the competencies detailing whether a competency or a quality is:

- Non-negotiable
- Very Important
- Important
- Negotiable
- Nice to have

The second column is for describing whether we are dealing with competency or a quality. The third column describes the quality or the competency in detail. The last column is the score rating column, which rates the competencies and qualities from 1–5, where:

- Non- negotiable (100%) = 5 (Five)
- Very Important (90%) = 4 (Four)
- Important (80%) = 3 (Three)
- Negotiable (70%) = 2 (Two)
- Nice to have (60%) = 1 (One)

The proposed FSECQ is presented as follows.

Table 3 below presents the proposed framework of evaluating the competencies and qualities of board of directors as a collective in an SOC. This framework is discussed in detail in Section 8.

**Table 3.** Proposed Framework of Evaluating the competencies and qualities of board of directors as a collective in an SOC.

| Category | Quality or Competence | Detailed Description | Score |
|---|---|---|---|
| Very important qualities competencies for collective directors | Competencies | Risk Management | **4** |
| Negotiable qualities directors | Competencies | Financial accounting, auditing and financial reporting background | **2** |
| | | Corporate Communications | |
| Nice to have collective directors | Competencies | Legal Background | **1** |
| | | Human Resources and Industrial Relations background | |
| | | Citizen-Centric | |

Source: Researcher's Conceptualisation.

## 8. The Evaluation Framework for the Proposed FSESQ

When screening a potential director's competencies, the process should be based on calculating the individual director score as follows.

### 8.1. Determining the Target Scores of Individual Director Competencies

From the ranking and rating process, 10 (ten) individual director competencies were derived. These competencies were ranked as 100% non-negotiable, 90% very important, and 80% important. Rating calculations were then conducted on these individual competencies, and the required score for competencies of individual directors were determined below.

100% Non-negotiable Competencies: Six (6) competencies of individual directors ranked 100%, being non-negotiable at this level. According to the proposed FSECQ, a potential board candidate must have all these six competencies as these qualities are non-negotiable. Consequently, if a director lacks even one of these competencies, they should not even be shortlisted for interviews, as these competencies are non-negotiable competencies

These competencies include educational qualification (in general), sound strategic communication skills, sound strategic interpersonal relations skills, understanding the organisation's purpose and its role in the country's development, good ethical standards, understanding the importance of managing public resources, and critical thinking.

90% Very Important Competencies: Only one competency of individual directors ranked at this level. This means that to qualify as a board member in a SOC potential board, it is very important to have an undergraduate degree. This means during the screening process, individuals with an undergraduate degree score higher than those without a junior degree.

80% Important competencies: Three (3) individual director competencies ranked at 80% level. This means a candidate should have educational qualifications (post-graduate), political acumen/awareness, and experience in other board positions to score high during the screening process.

### 8.2. Determining the Target Scores of Individual Director Qualities

Nine (9) individual director qualities were derived from the ranking and rating process. These qualities were ranked 100% non-negotiable, 90% very important, and 70% negotiable.

100% Non-Negotiable qualities: Seven (7) qualities of individual directors ranked 100% at this level. According to the proposed FSECQ, all potential board candidates must have all these seven qualities before being short-listed for interviews as potential board members in an SOC.

These non-negotiable qualities are ethical conduct and integrity; no tolerance for ethical violations; always fair, just, and unbiased; respect for fellow board members and executives; honest, reliable, and trustworthy; encourage executives and employees to thrive, flourish, and develop innovative ideas; and walk the talk of ethical values.

90% Very Important qualities: Only one individual director quality ranked at this level. This means for a potential board member to score high during the screening process, the potential board member should maintain high ethical standards.

70% Negotiable qualities: Like the above, only one individual director quality ranked at this level. This means that for a potential board member to score high in their screening process, they should be kind and act in a manner that is always beneficial to the board.

### 8.3. Determining the Target Scores of Board of Directors' Collective Competencies

From the ranking and rating process, seven (7) competencies of the board of directors were determined. These competencies ranked 90% very Important, 70% negotiable, and below 70% nice to have. Rating calculations were then conducted on these collective competencies and the required score for collective competencies of directors were determined as shown below.

90% Very Important Competencies: One (1) competency of the board of directors ranked 90% at this level. Based on the proposed FSECQ, the board of directors as a collective should have some board members with these skills for the board to be considered acceptable. This means for a board to be accepted as competent, it should have some members who have risk management skills.

70% Negotiable Competencies: Two collective competencies of the board of directors were ranked at this level. This means for a board to score high and be accepted as competent, it should have members who have financial accounting, auditing, and financial reporting background or some board members with corporate communication skills.

60%% Nice to have Competencies: Four (4) competencies of the board of directors ranked at this level.

This means for a board to score high and be accepted as competent, it would be nice to have board members with the following skills: legal background, human resources, industrial relations background, citizen-centric, and technologically savvy.

### 8.4. Guidelines on Screening the Potential Board Candidates

When an individual board member is being screened as a potential board candidate in an SOC, they must have the following non-negotiable competencies and qualities before they can even be considered for short listing for interviews:

Non-Negotiable Competencies

- Educational qualification (In General);
- Sound strategic communicating skills;
- Sound strategic interpersonal relations skills;
- Understanding the organisation's purpose and its role in the country's development;
- Good ethical standards and understanding the importance of managing public resources;
- Critical thinking.

Non-Negotiable qualities

- Ethical, and acting with integrity;
- No tolerance for ethical violations;
- Always fair, just, and not biased;
- Respect for fellow board members and executives;
- Honest, reliable, and trustworthy;
- Encourage executives and employees to thrive, flourish, and develop innovative ideas;
- Walk the talk of ethical values.

It is worth noting that if a board member does not possess even one of these competencies and qualities, they are automatically rejected as a potential candidate.

The second step is to calculate the scores of candidates with the above non-negotiable competencies and qualities. Those candidates that have generally scored high in the screening process are then be shortlisted.

### 8.5. Guidelines on Evaluating the Existing Board of Directors

Regarding the board of directors' competencies as a collective, no non-negotiable competencies or qualities are required. Instead, the board should collectively score high on one very important skill (being risk management skill), one important skill, and three negotiable skills for the existing board members to be retained and not be changed.

### 8.6. Examples Illustrating How to Score Individual Directors during the Screening Process

Whether a potential board candidate is recruited will depend on how they score as individuals on the FSECQ. The method for calculating the director's scores can be better be illustrated as shown in Table 4 below.

**Table 4.** Illustrative example showing how to score individual director competencies and qualities.

| | | The Description of the Competencies, Qualities and Screening and Evaluation Processes | Description Assigned | Total Score | Candidate 1: Score | Candidate 2: Score |
|---|---|---|---|---|---|---|
| **Individual Director Competencies** | 100% | Educational qualification (In General) | Non-Negotiable | 5 | 5 | 5 |
| | 100% | Sound strategic communicating Skills | Non-Negotiable | 5 | 5 | 5 |
| | 100% | Sound Strategic Interpersonal Relations skills | Non-Negotiable | 5 | 5 | 5 |
| | 100% | Understand the purpose of the organisation and its role in the country development | Non-Negotiable | 5 | 5 | 5 |
| | 100% | Good ethical standards and understand the importance of managing public resources | Non-Negotiable | 5 | 5 | 5 |
| | 100% | Critical Thinking | Non-Negotiable | 5 | 5 | 5 |
| | 90% | Educational qualification (Undergraduate) | Very Important | 4 | 4 | 0 |
| | 80% | Educational qualification (Post Graduate) | Important | 3 | 3 | 3 |
| | 80% | Political Acumen/Awareness | Important | 3 | 3 | 3 |
| | 80% | Experience in serving in other board positions | Important | 3 | 0 | 0 |
| | | Required competencies | | 43 | 40 | 36 |
| **Individual Directors Qualities** | 100% | Ethical, and acting with integrity | Non-Negotiable | 5 | 5 | 5 |
| | 100% | No tolerance for ethical violations | Non-Negotiable | 5 | 5 | 5 |
| | 100% | Always fair, just, and not biased | Non-Negotiable | 5 | 5 | 5 |
| | 100% | Respect for fellow board members and executives | Non-Negotiable | 5 | 5 | 5 |
| | 100% | Honest, reliable, and trustworthy | Non-Negotiable | 5 | 5 | 5 |
| | 100% | Encourage executives and employees to thrive, flourish, and develop innovative ideas | Non-Negotiable | 5 | 5 | 5 |
| | 100% | Walk the talk of ethical values | Non-Negotiable | 5 | 5 | 5 |
| | 90% | Regularly discuss high ethical standards | Very Important | 4 | 4 | 0 |
| | 70% | Kind and acting in a manner that is always beneficial to the team. | Negotiable | 2 | 0 | 0 |
| | | Required qualities | | 41 | 39 | 35 |
| | | Total required competencies and qualities | | 84 | 94% | 84% |

***Source:** Researcher's Conceptualisation.* NB: All shortlisted candidates must have all non-negotiable (100%) competencies and qualities. **Conclusions:** Candidate 1 seems to have most of the pre-required and highly ranked competencies and qualities and is, therefore, a better candidate to appoint.

Table 4 provides an example of scoring two potential SOC board candidates based on their competencies and qualities and how they compare to other candidates. The table also concludes on which candidate is likely to be chosen over the other based on the FSECQ. It is worth noting that no applicant will be shortlisted unless they have all the non-negotiable skills. Therefore, all candidates that proceed to the interview process should have these non-negotiable skills.

Furthermore, Table 5 also provides an example of two boards of directors and how the two boards score based on the skills combinations among the board of directors as a collective. The board with the highest score (i.e., the required skills) is the one that is then retained.

**Table 5.** Illustrative example showing how to score competencies and qualities of board of directors.

| | | The Description of the Competencies, Qualities | Description Assigned | Rating | Board 1 Skills Combination: Score | Board 2 Skills Combination: Score |
|---|---|---|---|---|---|---|
| **Board as a Collective** | 90% | Risk Management | Very Important | 4 | 4 | 0 |
| | 70% | Financial accounting, auditing, and financial reporting background | Negotiable | 3 | 3 | 3 |
| | 70% | Corporate Communications | Negotiable | 3 | 3 | 0 |
| | Below 70% | Legal Background | Nice to have | 2 | 0 | 2 |
| | Below 70% | Human Resources and Industrial Relations background | Nice to have | 2 | 2 | 2 |
| | Below 70% | Citizen-Centric | Nice to have | 2 | 0 | 2 |
| | Below 70% | Technologically savvy | Nice to have | 2 | 2 | 2 |
| | | | | 18 | 14 | 11 |
| | | | | | 78% | 61% |

*Source: Researcher's Conceptualisation.* **Conclusions:** Board 1 is more acceptable as it has most of the required set of skills.

Legend:

| | | |
|---|---|---|
| 100% | 5 | Non-Negotiable |
| 90% | 4 | Very Important |
| 80% | 3 | Important |
| 70% | 2 | Negotiable |
| Below 70% | 1 | Nice to have |

## 9. Conclusions

As noted in the discussion on the agency theory, the resource dependency theory, the human capital theory, social capital theory, and the trait theory, directors are key in the strategic operation of the SOC, which means directors' integrity, expertise, and competencies are crucial for the running of the organisation. This ideology is also confirmed by the discussions on the legislative and literature analysis of this study.

While the literature is varied and inconsistent, these discussions have also revealed that the existing theories and legislations are vague in describing the competencies and qualities of directors as well as on the methods that can be used to screen and evaluate directors. This demonstrates the need for a comprehensive framework that sets out the specific competencies and qualities required from SOC directors.

Therefore, this study has developed the FSECQ, which will assist ministers, policymakers, investors, the board of directors, and other stakeholders in screening and evaluating the competencies and ethical values of SOC board members.

**Author Contributions:** M.H. and T.M. were involved in the visualization, conceptualization, methodology, validation, and formal analysis of the research study. M.H. was involved in the investigation, writing—original draft preparation, writing—review and editing. Supervision, was done by T.M. and B.M. All authors have read and agreed to the published version of the manuscript.

**Funding:** This research received no funding.

**Data Availability Statement:** The data presented in this study are available on request from the corresponding author. The data are not publicly available due to confidentiality promised to participants when survey were conducted.

**Conflicts of Interest:** The authors declare no conflict of interest.

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
