# Peer review of "Framework for Screening and Evaluating the Competencies and Qualities of the Board of Directors in South Africa’s State-Owned Companies"

_jrfm, doi:10.3390/jrfm15110492_

Round 1
Reviewer 1 Report
- The abstract needs to present in a clear manner.
- What are the theoretical and practical contributions?
- The conclusion statement is too short and does not support the study results and purposes.
- The results needed to presents in a professional manner with more discussion and interpretation.
- Itis not preferable to state points within the introduction statement.
- What are the practical and theoretical implications of the study findings ?
- The overall content of the manuscript does not present in a systematic and clear manner.
Author Response
Dear Reviewer
Thank you for your review notes. I have attached your points and my response.
Point 1: The abstract needs to present in a clear manner.
Response 1: The abstract has been reworded and more information has been provided to make it more clear.
Point 2: What are the theoretical and practical contributions?
Response 2: The primary theories that have been adopted in this study are agency theory, resource dependency theory, human capital theory, social capital theory and trait theory, as these theories are associated with the competencies and qualities of directors and they have been discussed in detail.
Point 3: The conclusion statement is too short and does not support the study results and purposes.
Response 3: The conclusion statement has been revised and more details have been provided
Point 4: The results needed to present in a professional manner with more discussion and interpretation.
Response 4: Discussion and interpretation of results have been attended to.
Point 5: It is not preferable to state points within the introduction statement.
Response 5: Points discussed in the introduction chapter have been taken out and replaced with a paragraph discussion.
Point 6: What are the practical and theoretical implications of the study findings?
Response 6: The practical and theoretical implications of the study have been discussed in detailed following the point raised.
Point 7: The overall content of the manuscript does not present in a systematic and clear manner.
Response 7: The manuscript has been revised and more detail has been provided as requested by the reviewer.
Reviewer 2 Report
The paper has potential for publication, however, the paper structure needs to be significantly improved.
1. The authors need to improve their paper from the introduction to the conclusion.
2. There is a need to clearly highlight the literature gap and position the paper in relation to the existing literature. The authors need to complement their literature section with recent studies (e,g Mbanyele 2022).
3. The authors need to highlight the economic theory linked to their paper (The are some classical papers on agency theory below).
4. The conclusion is too shallow. There is a need to highlight some policy recommendations.
Suggested literature to add to the paper
Fama, E. F., & Jensen, M. C. (1983). Agency problems and residual claims. Journal of Law and Economics, 26, 327–349.
Mbanyele, W. 2020. Do Busy Directors Impede or Spur Bank Performance and Bank Risks? Event Study Evidence From Brazil, SAGE Open 10(2):1-17.
Jensen, M. C., & Meckling, W. H. (1976). Theory of the firm: Managerial behavior, agency cost and ownership structure. Journal of Financial Economics, 3, 305–360
Author Response
Dear Reviewer
Thank you very much for your review notes, I have responded to all of them in an attachment.

Round 2
Reviewer 1 Report
Thank you for considering the comments and suggestions, now I can recommend the manuscript for publication
Reviewer 2 Report
The paper is now fine.